# Cutaneous Epithelioid Angiomatous Nodule: Report of a New Case and Literature Review

**Margaux Dubus** [1,2] **and Jean Kanitakis** [2,3,*]

1. Department of Dermatology, University of Grenoble Alpes, 38700 La Tronche, France
2. Department of Dermatology, Ed. Herriot Hospital Group, University of Lyon, 69003 Lyon, France
3. Department of Pathology, Centre Hospitalier Lyon Sud, 69310 Pierre Bénite, France
* Correspondence: jean.kanitakis@univ-lyon1.fr

**Abstract:** Cutaneous epithelioid angiomatous nodule is a rare benign vascular tumour of the skin with characteristic microscopic features, of which 65 cases have so far been reported after the initial description of this entity in 2004. We present here a new typical case of this rare lesion and provide a comprehensive review of all the previously published cases, delineating the salient clinicopathological features of this rare tumour.

**Keywords:** cutaneous epithelioid angiomatous nodule; epithelioid vascular proliferations

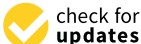



## 1. Introduction

Cutaneous epithelioid angiomatous nodule (CEAN) is a rare benign vascular tumour of the skin first described by Brenn and Fletcher in 2004 [1], of which 65 cases have so far been reported [1–29]. It belongs to the spectrum of epithelioid vascular proliferative lesions, including other neoplasms, either benign (such as epithelioid haemangioma and pyogenic granuloma) or malignant (such as epithelioid haemangioendothelioma and epithelioid angiosarcoma). We report here a new typical case of CEAN and present a thorough review of the relevant literature in order to outline the main clinicopathological features of this rare entity and discuss the differential diagnosis with other epithelioid proliferations.

## 2. Case Report

A 46-year-old man presented for an asymptomatic, slowly growing erythematous nodule of about five months' duration, on the right side of the back. There was no history of prior erosion, ulceration or systemic complaints. His medical history included pulmonary sarcoidosis, but he was not receiving treatment. Physical examination revealed a 6 mm well-circumscribed, erythematous nodule (Figure 1). The lesion was completely excised surgically under local anaesthesia with the suspected clinical diagnosis of vascular neoplasm (possibly thrombosed haemangioma).

Histological examination of routinely-stained sections showed a well-circumscribed exo-endophytic dermal nodular lesion. The overlying epidermis was slightly thinned in the centre and slightly hyperplastic laterally, forming a collarette (Figure 2). The lesion involved the upper- and mid-dermis but spared the deep dermis and the subcutaneous adipose tissue; it was composed of a proliferation of blood capillary vessels with more or less thick walls, containing large endothelial cells with ample, eosinophilic, glassy cytoplasm (Figure 3). With an increasing depth, the proliferating vessels had larger lumina and were filled with red blood cells. The surrounding dermis contained several extravasated red blood cells, hemosiderin deposits and a population of large, round or ovoid cells with an epithelioid appearance, containing ample, eosinophilic glassy cytoplasm, and large, centrally-located, occasionally bilobated nuclei with conspicuous nucleoli. Some of these cells contained red blood cells within cytoplasmic vacuoles, representing (abortive) endothelial lumina (Figure 4). Rare mitoses were seen (Figure 5) but were not atypical.

Immunohistochemically, the endothelial cells of the proliferative blood vessels expressed the endothelial antigens CD31, CD34 and ERG (Figures 6–8) but were negative for keratins (AE1/AE3) (Figure 9), hormonal receptors (for estrogen, androgen and progesterone) and the late nuclear antigen of the HHV-8. Immunolabelling for podoplanin (D2-40) revealed some small lymphatic vessels within the lesion, but the majority of the proliferative vessels were negative. The extravascular epithelioid cells expressed a similar phenotype with the endothelial cells lining vascular lumina, although the expression of endothelial markers was weaker. The proliferation antigen MIB-1 (Ki-67) was expressed by ca. 15–20% of the endothelial cells (Figure 10). Some CD68+ and/or CD163+ histiocytic cells were scattered within the lesion. Based on these findings, the diagnosis of CEAN was made.

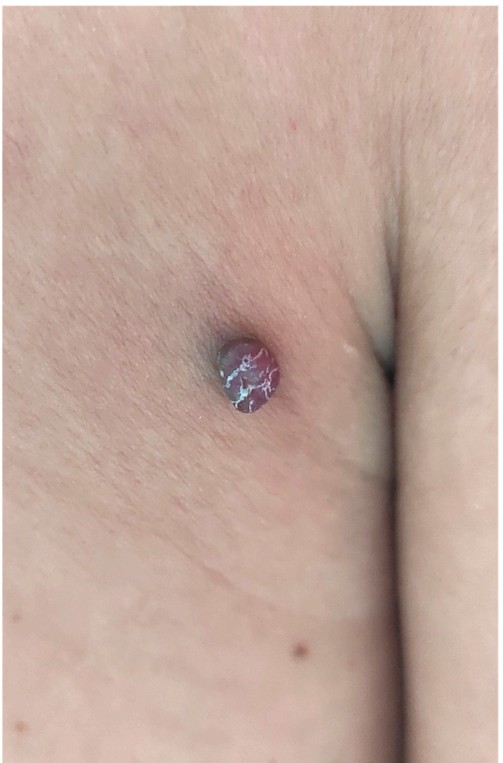

**Figure 1.** Erythematous nodular lesion on the back.

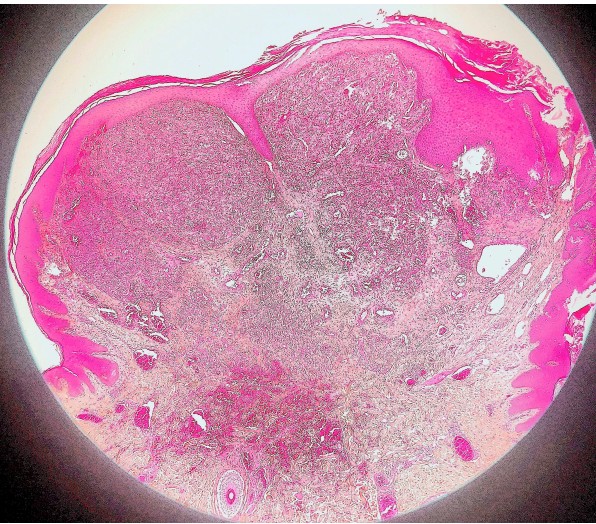

**Figure 2.** Scanning view of the lesion: well-demarcated, dermal nodule, surrounded by an epidermal collarette (haematoxylin–eosin–saffron stain).

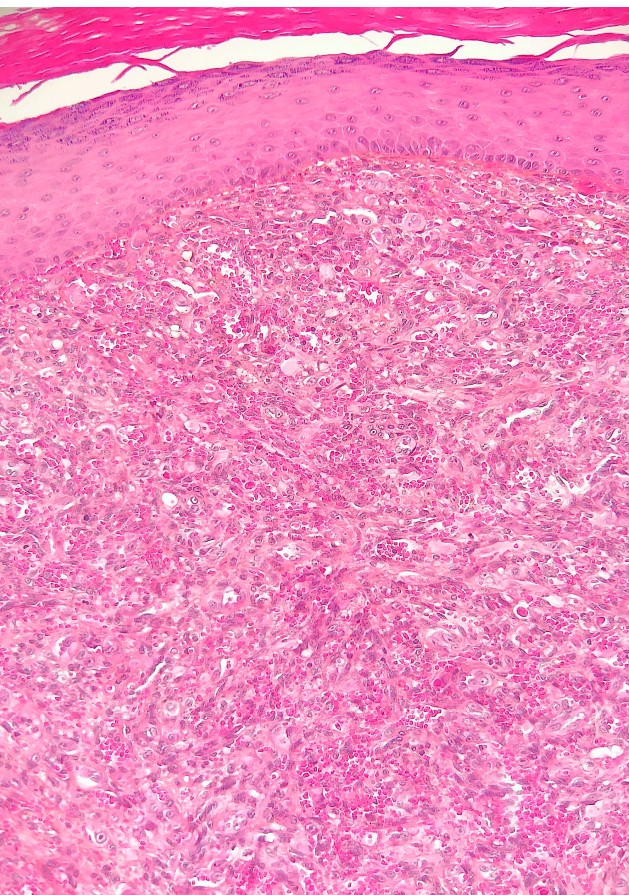

**Figure 3.** The lesion is composed of a proliferation of thick-walled blood capillary vessels containing large endothelial cells with ample, eosinophilic, glassy cytoplasm (haematoxylin–eosin–saffron stain).

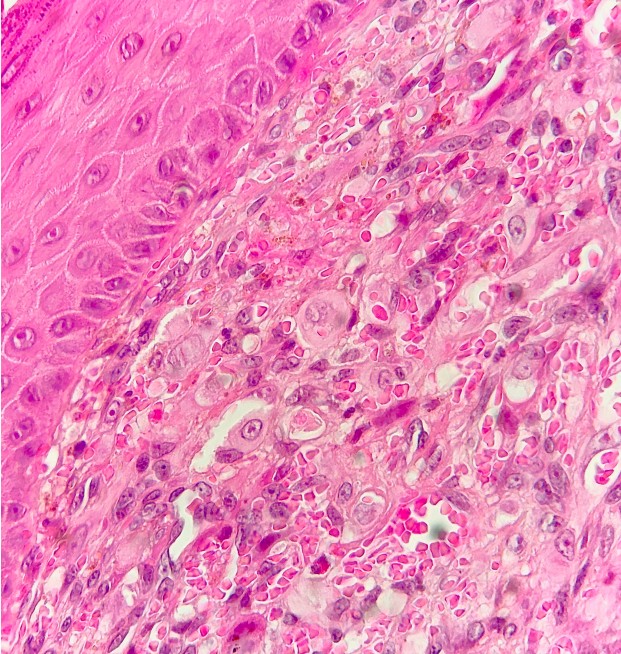

**Figure 4.** The characteristic epithelioid cells with ample, eosinophilic cytoplasm, and large, centrally located nuclei with conspicuous nucleoli are well visible; some of them contain red blood cells within intracytoplasmic lumina. Note the numerous extravasated red blood cells and haemosiderin deposits (haematoxylin–eosin–saffron stain, higher magnification of Figure 3).

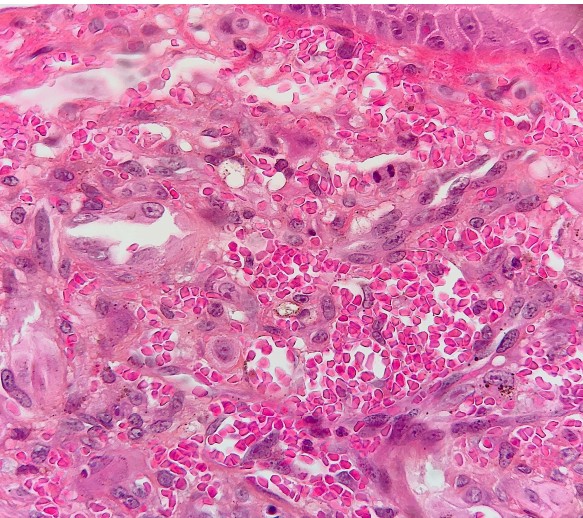

**Figure 5.** Occasional (normal) mitoses are seen (haematoxylin–eosin–saffron stain). Note also epithelioid endothelial cells lining vascular lumina.

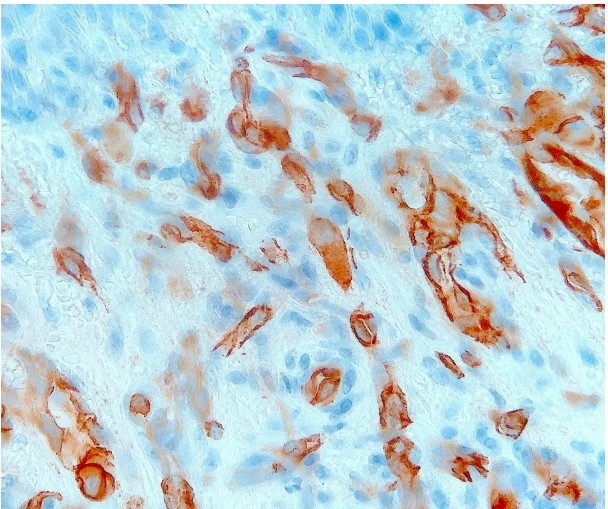

**Figure 6.** The tumour cells express CD31 (immunoperoxidase revealed with diaminobenzidine).

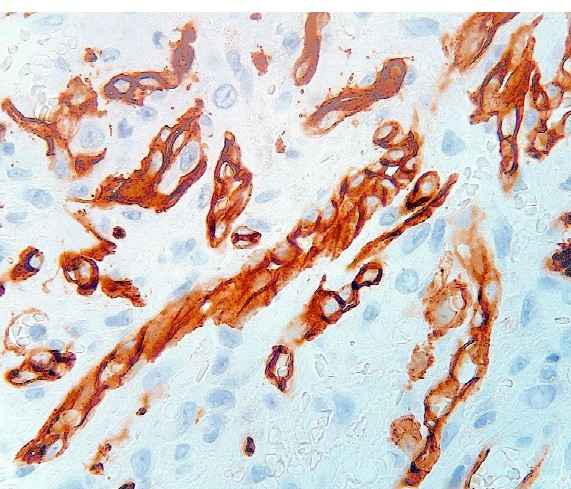

**Figure 7.** The tumour cells express CD34 (immunoperoxidase revealed with diaminobenzidine).

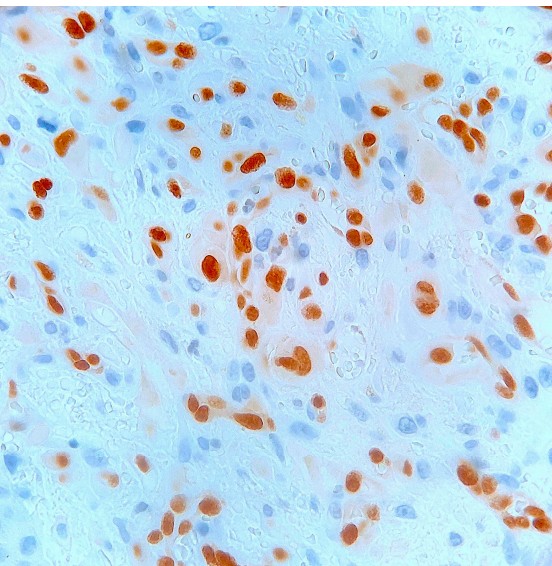

**Figure 8.** The tumour cells express nuclear reactivity for ERG (immunoperoxidase revealed with diaminobenzidine).

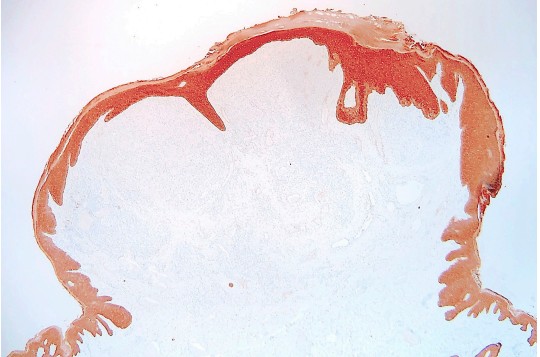

**Figure 9.** The tumour cells do not express keratin, contrasting with the overlying epidermis (immunoperoxidase revealed with diaminobenzidine).

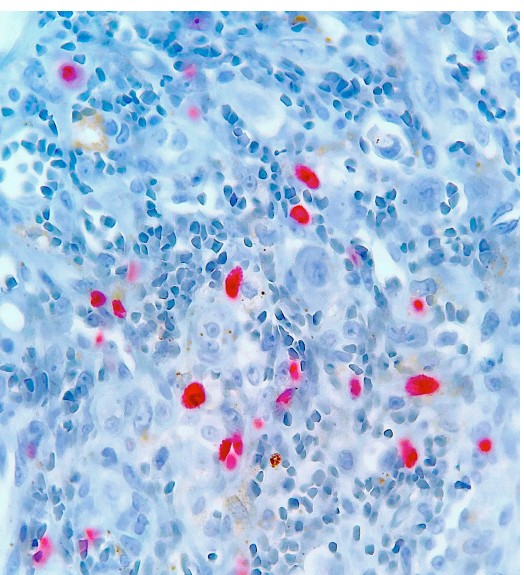

**Figure 10.** Occasional tumour cells express the MIB-1/Ki67 proliferation antigen (immuno-alkaline phosphatase revealed with fast-red).

Following excision of the lesion, no recurrence, metastasis or evidence of infection were observed after a follow-up of seven months.

## 3. Discussion

CEAN is a rare vascular proliferative lesion, of which 66 cases have so far been reported after the initial description of this entity in 2004 [1–29], including the case presented here. Clinically, CEAN presents as a bluish-red papule or nodule measuring 0.2–2.5 cm. The age of patients ranges from 13 to 84 years (mean 42 years). There is a slight, but probably non-significant, male predominance (37 men/29 women). The lesion was single in the majority of cases (53/66, 80%); multiple lesions were present in 20% of cases. Four patients had multiple lesions on several body sites [3,11,21,22]. CEAN develops on the head and neck (41%—including the face, nasal cavity, scalp, oral cavity, external ear), the trunk (38%) and the extremities (26%). It is usually asymptomatic, although some patients reported pain ($n = 9$), bleeding ($n = 4$) or pruritus ($n = 3$). CEAN usually develops over a short period of time (one week to 6 months), but seven patients had lesions present for 1–2 years, and in one patient the lesion had been present more than 30 years [19]. Most patients had no medical problems and were not taking any treatment. Five patients had a significant medical history for breast cancer [19], haemodialysis [4], nephrotic syndrome treated by cyclosporine [5], immunosuppressive treatment (without more information) [6] and pulmonary sarcoidosis (this case); however, these associations could be fortuitous. Rarely, CEAN coexists with other vascular lesions, including capillary malformations [7,25], pyogenic granuloma [21] and hepatic haemangiomas [24].

Microscopically, CAEN is usually localized in the superficial and mid-dermis, although it may also involve the deep dermis. It is as a rule (91%) well-circumscribed, albeit non-encapsulated. It manifests with a proliferation of blood vessels admixed with large epithelioid endothelial cells containing large nuclei and prominent nucleoli, and an abundant eosinophilic cytoplasm, often containing intracytoplasmic vacuoles. One case contained giant cells [6]. CAEN shows no signs of cytologic atypia or pleomorphism. Occasional mitoses are present in two-thirds of cases, but they are not abnormal, as in our case. Numerous (typical) mitoses were present in two cases (6, 24). Open vascular channels within the lesion were present in the majority of cases, in variable numbers, sizes, and shapes. They were only focal in 31 cases and with a homogenous distribution in 3 cases [24]. The epithelioid endothelial cells often contain intracytoplasmic vacuoles, representing (abortive) lumina within which red blood cells may be found ([3,4,6,12,14,18,19,21–23,26], this case). A chronic inflammatory infiltrate is almost consistently present, composed mainly of lymphocytes. Eosinophils, although in small numbers, are often scattered throughout the lesion. The adjacent dermis shows mild fibrosis (27%), haemosiderin deposits (21%) and dilated vessels (24%). A case with myxoid changes has been reported [10]. The epidermis overlying the tumour is often thickened, acanthotic and hyperkeratotic. Six out of the 66 cases (9%) showed ulceration or epidermal collarette formation (including the one reported here). By immunohistochemistry, the epithelioid endothelial cells almost invariably express the vascular markers CD31, CD34 and factor-VIII-related antigen. In our case the interstitial epithelioid cells expressed these vascular markers more weakly compared with the endothelial cells lining vascular lumina, suggesting a lower degree of differentiation. Staining for smooth muscle actin highlights the presence of pericytes ($n = 16/16$). Immunostaining for keratins ($n = 27$) and S-100 protein ($n = 17$) is negative. One case showed weak but diffuse membranous expression of EMA ($n = 1/10$), along with estrogen receptor expression, in over 80% of the epithelioid cells [19]. Our case was negative for hormonal receptors. Podoplanin (D2-40) is not usually expressed by the endothelial cells; it was present only in peripheral (lymphatic) vessels in three cases (including our own), negative in eight cases and diffuse and strong only in one case [19]. None of the 27 cases tested (including the one reported here) were immunoreactive for HHV-8. Ki-67 is expressed by fewer than 30% of tumour cells ($n = 6/6$).

The differential diagnosis of CEAN includes other vascular proliferations with an epithelioid appearance. Epithelioid angiosarcoma is larger, not well-circumscribed, involves the deep dermis and shows an infiltrative growth pattern along with cytologic features of malignancy, such as nuclear pleomorphism and numerous, atypical mitoses. In epithelioid haemangioendothelioma, the cells are arranged in cords within a chondromyxoid stroma. Epithelioid haemangioma is multinodular and contains more vessels lined by endothelial cells [30].

The aetiology of CEAN is not well known. The few associations with other systemic diseases [4,5,19] or vascular proliferations [7,21,24,25] could be fortuitous. There is no evidence favouring an infectious aetiology, contrary to other vasoformative proliferations such as Kaposi sarcoma (HHV-8) and bacillary angiomatosis (bartonellosis). CEAN is believed to be a reactive process, but the initiating trigger remains unknown. Contrary to pyogenic granuloma, trauma does not seem to be involved in the development of CEAN.

CAEN is usually treated with surgical excision. Some lesions regressed with topical corticosteroids [24]. One patient with multiple lesions was treated with topical and intralesional steroids (reported as ineffective) and cryotherapy with acceptable results and no recurrence after two sessions [8]. In another patient with multiple CEAN, the lesions gradually disappeared after ciclosporin discontinuation [5].

CEAN is a benign lesion, although it may recur following excision (8% of cases) [3,6]. In one case CEAN recurred at a new location [22]. However, metastases have never been reported, with the follow-up period ranging from 1 month to 7 years.

## 4. Conclusions

We report a new case of CEAN, a rare benign epithelioid vascular proliferation, and review the relevant literature. Histopathological examination is the mainstay of diagnosis, often aided by immunohistochemical staining. CEAN has distinctive histological features that allow separation from other epithelioid vascular tumours.

**Author Contributions:** Conceptualization, J.K.; methodology, M.D. and J.K.; validation, M.D. and J.K.; investigation, M.D. and J.K.; data curation, M.D. and J.K.; writing—original draft preparation, M.D.; writing—review and editing, M.D., J.K.; supervision, J.K. All authors have read and agreed to the published version of the manuscript.

**Funding:** This research received no external funding.

**Institutional Review Board Statement:** Not applicable.

**Informed Consent Statement:** Written informed consent has been obtained from the patient to publish this paper.

**Data Availability Statement:** The data presented in this study are available on request from the corresponding author. The data are not publicly available due to privacy reasons.

**Conflicts of Interest:** The authors declare no conflict of interest.

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
