# Peer review of "Cutaneous Epithelioid Angiomatous Nodule: Report of a New Case and Literature Review"

_dermatopathology, doi:10.3390/dermatopathology10010017_

Round 1

Reviewer 1 Report

None

Author Response

Thank you for your reviewing - no comments submitted, therefore we have no replies. 

Reviewer 2 Report

In this article, the authors report a new, typical case of CEAN. Detailed immunohistochemical studies were performed and the description is careful and convincing. The authors then analyzed 66 cases of CEAN, including their own case, and extracted several clinical and histopathological features of CEAN. The paper itself is an excellent review of CEAN, but I have a few minor comments, explained below.

Minor comments.

1. Microscopically, the case presented here showed an epidermal collarette.  Is collarette formation a common histopathological feature of CEANs?

2. Red material is seen in the cytoplasm of the large epithelioid cell located in the center of Figure 4. Is this an image of phagocytosis of an erythrocyte? Is such phagocytosis common in this tumor?

Author Response

Thank you for your kind comments. Please find below our replies tour your queries for minor revision:

  1. As already stated in the 'discussion', an epidermal collarette was seen in 6/66 cases of CEAN (9%), therefore this finding is not very common, although not exceptional.
  2. Indeed, occasional red blood cells can be seen in the cytoplasm of epithelioid cells. This is likely not due to phagocytosis, but corresponds to the presence of (abortive) intracytoplasmic lumina, as has already been reported in previous cases of Cutaneous Epithelioid Angiomatous Nodule 'see: Samal et al. Indian Dermatol Online J 2019;10:463). This finding was added in the revised version of our article (in the description of the case and the 'discussion'). Furthermore fig. 4 was substituted by another one, showing 2 epithelioid cells containing red blood cells. The legend of fig. 4 was modified accordingly.